# Self-Assembled Sandwich-like Mixed Matrix Membrane of Defective Zr-MOF for Efficient Gas Separation

**DOI:** 10.3390/nano15040279

**Published:** 2025-02-12

**Authors:** Yuning Li, Xinya Wang, Weiqiu Huang, Xufei Li, Ping Xia, Xiaochi Xu, Fangrui Feng

**Affiliations:** 1Jiangsu Key Laboratory of Oil-Gas & New-Energy and Transportation Technology, Changzhou University, Changzhou 213164, China; s22040857019@smail.cczu.edu.cn (Y.L.); li_optimism@163.com (X.W.); lixf@cczu.edu.cn (X.L.); s23040857055@smail.cczu.edu.cn (X.X.); s23040857058@smail.cczu.edu.cn (F.F.); 2Engineering Technology Research Center for Oil Vapor Recovery, School of Petroleum and Natural Gas Engineering, Changzhou University, Changzhou 213164, China; 3Department of Research and Development, Changzhou First Hydrocarbon Environmental Protection Sci-Tech Co., Ltd., Changzhou 213164, China; xia.ping.518@163.com

**Keywords:** gas separation, mixed matrix membrane, Zr-MOF, synergistic modification

## Abstract

Membrane technology has been widely used in industrial CO_2_ capturing, gas purification and gas separation, arousing attention due to its advantages of high efficiency, energy saving and environmental protection. In the context of reducing global carbon emissions and combating climate change, it is particularly important to capture and separate greenhouse gasses such as CO_2_. Zr-MOF can be used as a multi-dimensional modification on the polymer membrane to prepare self-assembled MOF-based mixed matrix membranes (MMMs), aiming at the problem of weak adhesion or bonding force between the separation layer and the porous carrier. When defective UiO-66 is applied to PVDF membrane as a functional layer, the CO_2_ separation performance of the PVDF membrane is significantly improved. TUT-UiO-3-TTN@PVDF has a CO_2_ permeation flux of 14,294 GPU and a selectivity of 27 for CO_2_/N_2_ and 18 for CO_2_/CH_4_, respectively. The CO_2_ permeability and selectivity of the membrane exhibited change after 40 h of continuous operation, significantly improving the gas separation performance and showing exceptional stability for large-scale applications.

## 1. Introduction

Petrochemical production, coal gasification, fossil fuel combustion and other industrial activities emit CO_2_, CH_4_ and other waste gasses, which cause energy waste and economic losses, further aggravate haze weather and prompt the deterioration of the greenhouse effect [1]. Gas separation technology plays a key role in promoting energy efficiency and reducing carbon emissions [2,3,4]. Membrane-based separation technology can reduce energy consumption by up to 90% compared to heat-driven distillation methods, which have shown great prospects after several technological breakthroughs [5]. Excellent membrane materials (such as thin film composite membranes) should have suitable pore sizes and transport channels that ensure selective penetration [6,7,8]. In addition, the membrane should also have high porosity and thin thickness to minimize mass transfer resistance to achieve high-throughput separation [9]. Consequently, it has found extensive applications, such as oily wastewater treatment [10], water purification [11], food processing [12], gas separation [13] and other practical industrial applications [14].

Due to the high selectivity and large pore volume, MOFs have aroused great interest in the field of gas adsorption and separation [15,16]. They are considered one of the most important materials for efficient gas separation [17,18,19]. MOF-based mixed matrix membranes (MMMs) can use the adjustable function of MOFs to meet different gas separation requirements and the “trade-off” between gas selectivity and permeability [20,21,22]. However, the limited interaction between MOFs’ filler and polymer matrix leads to non-selective voids [23], increased rigidity and pore plugging, which become technical difficulties for MMMs [24,25,26].

Some research progress has been made in this field to improve the separation performance [27]. Qiu et al. developed a novel thin film nanocomposite (TFN) membrane based on PIM-1 by incorporating nanosized UiO-66-NH_2_ and carboxylate PIM-1 (cPIM-1) as composite fillers [28]. The UiO-66-NH_2_/cPIM-1 particles integrate with the polymer chains to form a stable 3D network, effectively preventing chain relaxation and physical aging, thereby significantly enhancing the gas separation performance. Shen et al. fabricated a UiO-66-NH_2_-PEBA MMM, which demonstrated excellent and stable CO_2_/N_2_ separation performance under humid conditions, achieving a CO_2_ permeability of 130 Barrer and a CO_2_/N_2_ selectivity of 72, surpassing the upper bound of conventional polymer membranes [29]. In addition, Pebax-based composite hollow fiber membranes are also used for industrial gas separation applications. Hou et al. utilized UiO-66 to fabricate a nanocomposite hollow fiber membrane, and UiO-66 was incorporated into the thin selective layer of Pebax, resulting in simultaneous improvements in both CO_2_ permeance and selectivity [30]. High-performance membrane materials can be synthesized by different preparation strategies, and the membrane with multilayer structure should be studied emphatically [31]. However, the concurrent optimization of high permeability and high selectivity remains a central challenge in the field of polymer membranes for gas separation [32].

In this work, the membrane coated with metal–phenolic (TTN) was constructed by a stepping assembly, alternating complexation and cooperative coordination. A self-assembled membrane featuring a sandwich-structured selective layer has been successfully prepared. The MOF layer is fixed through the coordination crosslinking between Ti ion and polyphenol tannic acid (TA), which effectively improves the gas separation performance. The innovative design of this multi-layered structure not only enables the thickness to be precisely controlled by different types of MOFs and the assembly concentration but also improves the separation efficiency of complex gas mixtures through optimization of the selective layer.

## 2. Materials and Methods

### 2.1. Materials

Polyvinylidene fluoride (PVDF, the polymer precursor, 950 kDa), Zirconium chloride (ZrCl_4_, 99.9%), tannic acid (TA, 98%), acetic acid, Dihydroxybis (ammonium lactato) and Titanium (IV) (Ti-BALDH, 15%) were supplied by Sinopharm Chemical Reagent Co., Ltd. (Shanghai, China). Additionally, 2-Aminoterephthalic acid (NH_2_-BDC, 98%) was purchased from Aladdin Biochemical Technology Co., Ltd. (Shanghai, China). Other chemicals were procured from Changzhou Runyou Trade Co., Ltd. (Changzhou, China), including polyvinylpyrrolidone (PVP, K30, 40,000–60,000 Da), ethanol and N,N-dimethylacetamide (DMF, ≥99.5%). The gasses used in the experiments (CH_4_, N_2_, CO_2_, 99.99%) were supplied by Changzhou Huayang Gas Co., Ltd. (Changzhou, China). All experiments utilized deionized water. All reagents used in the experiments were used without further purification.

### 2.2. Synthesis of UiO-66-NH_2_

UiO-66-NH_2_ was synthesized using a solvothermal method, as reported in the literature. ZrCl_4_ (1 mmol) was added to 30 mL of DMF, and NH_2_-BDC (1 mmol) was added to 20 mL of DMF, with solutions being ultrasonically stirred until dissolving and then transferred into a high-pressure reactor for 120 °C for 24 h. After natural cooling to room temperature, the resulting white solid was separated by centrifugation with 1000 rpm, washed with 60 mL of ethanol twice at 70 °C for 1 h, dried under vacuum at 150 °C for 12 h and stored for further use.

### 2.3. Synthesis of Defective UiO-66-NH_2_

ZrCl_4_ (1 mmol) and NH_2_-BDC (1 mmol) were dissolved in 30 mL of DMF and stirred with 1000 rpm at room temperature for 10 min. Acetic acid was added to the mixed solution at a fixed molar ratio (UiO-66-NH_2_:acetic acid = 1:15, 30, 70, 100) and transferred into a high-pressure reactor for 120 °C for 24 h to induce the material to form more defect sites. After natural cooling to room temperature, the resulting white solid was separated by centrifugation with 1000 rpm, washed with 60 mL ethanol twice for 1 h to remove impurities, and dried. According to the different molar ratio of acetic acid, d-UiO-1 (1:15), d-UiO-2 (1:30), d-UiO-3 (1:70) and d-UiO-4 (1:100) were obtained.

### 2.4. Synthesis of d-UiO-66-NH_2_ with Thin-Film Nanocomposite (TFN)

As shown in Figure 1, TFN with d-UiO-66-NH_2_ is formed by the alternating assembly of metal–phenolic coordination active coating (TTN multilayer) and UiO nanocrystals. Before assembly, the membrane is washed twice with 30 mL methanol for 1 h and stored for at least 12 h to remove the impurities at room temperature. The PVDF membrane [33,34] was immersed in a 5% tannic acid (TA) methanol solution for 4 h to achieve sufficient adsorption of TA [35]. After that, the membrane was transferred to a Ti-BALDH with a methanol solution and impregnated for 1 h to assemble the TTN primer coating (provides space for growing UiO nanocrystals). TTN@PVDF was immersed in d-UiO-66-NH_2_ dispersed methanol solution for 1 h to obtain TUT-UiO@PVDF. We repeated the above steps to obtain TUT-UIO-TTN@PVDF. After preparation, the membrane was immersed in Tris-HCl buffer (pH = 8) for 0.5 h and named as TUT-UiO-1-TTN@PVDF, TUT-UiO-2-TTN@PVDF, TUT-UiO-3-TTN@PVDF and TUT-UiO-4-TTN@PVDF. The innovative design of the composite sandwich-like structure consists of a TUT, UiO and TTN layer.

### 2.5. Characterization

The morphology of TUT-UiO-TTN@PVDF was observed using a scanning electron microscope (SEM, SUPRA-55, Zeiss, Oberkochen, Germany). An EDS (Energy Dispersive Spectroscopy) analysis was conducted on the membrane samples. Cross-sectional samples were prepared using a liquid nitrogen fracture method. Prior to observation, all samples were sputter-coated with gold to render them conductive. The chemical composition of the samples was analyzed using Fourier transform infrared spectroscopy (FT-IR, Thermo Fisher, Waltham, MA, USA, */IS50) and X-ray photoelectron spectroscopy (XPS, Thermo ESCELAB 250XI, Thermo Fisher Scientific, Waltham, MA, USA). The crystalline structure was studied using X-ray diffraction (XRD, MAX2500, Rigku, λ = 0.15406 nm, collection range from 2° to 80°). The mechanical properties of the membranes were tested using a universal testing machine (AGS-10KND). The thermal stability of the composites was studied using a thermogravimetric analyzer (TGA/DSC 3+, Mettler Toledo). The structural properties of the samples were characterized using a Quantachrome Autosorb iQ2 analyzer. The specific surface area (*S*_BET_)was calculated using the Brunauer–Emmett–Teller (BET) equation. The total pore volume was measured by N_2_ adsorption at P/P_0_ = 0.995. The micropore volume and micropore specific surface area *(S*_BET_) were estimated using the t-plot method. Pore size distribution (PSD) was determined using the non-local density functional theory (NLDFT) model. The gas permeability coefficient was determined by constant pressure and a variable volume method [36]. The sample is fixed, with the gas passing through with a test temperature and pressure of 25 °C and 100 kPa. Repeated measurements are taken three times to take the average, and the gas flow is measured by the flowmeter.

## 3. Results and Discussion

### 3.1. Characterization of TUT-UiO-TTN@PVDF Membranes

The SEM images in Figure 2 clearly show the structure of the membrane under an operating voltage of 5 kV. Compared with the porous surface of the original PVDF membrane (Figure 2a,e), the assembly of the Ti-UiO-TA (TUT) functional layer makes the membrane surface rougher and denser. As shown in Figure 2f, d-UiO is uniformly dispersed on the membrane surface to form a layer with a thickness of about 100 nm. Meanwhile, the d-UiO-3@TUT-PVDF membrane surface was scanned by EDS to verify the loading of d-UiO (Figure 2j). Unlike PVDF membranes, d-UiO nanocrystals with an average diameter of 20–50 nm are uniformly distributed along the membrane surface without accumulation. Moreover, due to the defective active sites of d-UiO increasing, more d-UiO become involved in the formation of the functional layer, increasing the microporous structure of the material, and more unsaturated metal adsorption sites are formed. However, the d-UiO is distributed unevenly in the membrane cavity and has a lower loading without a TTN-layer modification. It is concluded that the complete mixed functional layer of the TUT-UiO-TTN@PVDF membrane can be constructed through the alternating assembly and complexation of Titanium–tannic acid (TTA) and d-UiO.

In addition, the SEM images of UiO-66-NH_2_ and d-UiO-3 verified the modification of the defects [37]. As can be seen from Figure 2i, original UiO-66-NH_2_ exhibits a small grain symbiotic aggregate of between 10 and 20 nm. After acetic acid regulation, d-UiO-3 became more regular and increased in grain size with a uniform and smooth surface, with acetic acid acting as a regulator to promote the nucleation process of Zr_6_O_4_(OH)_4_ during the synthesis of Zr-MOF while accelerating the crystal growth of Zr-MOF.

As shown in Figure 3a and Table 1, the micropore of d-UiO-4 is higher than that of UiO-66-NH_2_ with a higher *S*_BET_. The functional layer was deposited not only on the membrane surface but also on the internal pore surface of the membrane; the porosity of theTUT-UiO-TTN@PVDF membrane was reduced compared with the original PVDF membrane. In order to characterize the structure of d-UiO in the functional layer of TUT-UiO-TTN@PVDF, the *S*_BET_ of UiO-66-NH_2_ and d-UiO-4 are compared in Figure 3b. It can be seen that the adsorption isotherms of d-UiO-4 and UiO-66-NH_2_ both show typical type-I isotherms, which proves that d-UiO-4 is a highly microporous material with narrow pore size distribution. As the pressure increases (P/P_0_ > 0.95), the adsorption curve increases sharply, indicating that the capillary condensation phenomenon occurs due to the aggregation of nano defects. Compared with UiO-66-NH_2_ (671 m^2^·g^−1^, 0.25 cm^3^·g^−1^), the *S*_BET_ of d-UiO-4 (1048 m^2^·g^−1^, 0.40 cm^3^·g^−1^) increased by 56%, which is due to appropriate carboxylate etching with acetic acid-produced ligand defects and additional cavities that improved the micropore structure. Furthermore, more adsorption sites are formed, breaking the limitation of a single micropore. In addition, the non-local density functional theory (NLDFT) was used to further study the pore distribution of UiO-66-NH_2_ and d-UiO-4, as shown in Figure 3c.

### 3.2. Crystal Structure of TUT-UiO-TTN@PVDF Membranes

The chemical structure of TUT-UiO-X-TTN@PVDF was monitored by XRD, FT-IR, and XPS spectra (Figure 4). The modified d-UiO-4 and UiO-66 both show crystal faces (111), (002) and (600) at 2θ = 7.2°, 8.4° and 25.6°, indicating that d-UiO-4 and UiO-66 have similar structures. However, the crystallization peak strength of d-UiO-4 is higher than UiO-66, demonstrating that d-UiO-4 has better crystallinity. After alternate assembly, both TUT-UiO-4-TTN@PVDF and the original PVDF membranes contain α phases (18.3°, 26.4°) and β phases (20.1°) of PVDF (Figure 4b). Compared with the original PVDF membrane, TUT-UiO-4-TTN@PVDF has reduced peak strengths at 18.3° α (020), 26.4° α (021) and 20.1° β (110/200), which are attributed to the surface etching of the PVDF membrane by the weakly acidic phenolic compound of TA [38]. The higher the degree of crystallinity, the higher the tensile strength, elastic modulus and hardness of the film generally; this indicates that the molecular chains in the crystalline region are arranged neatly and tightly and that the intermolecular forces are enhanced so that the material can better resist deformation when it is stressed.

The alternate assembly strategy of the TTN reduces the curves of the XRD, while the additional characteristic peaks are 7.2° and 8.4°, indicating the successful loading of d-UiO-4 on the PVDF membrane. Similarly, TUT-UiO-4-TTN@PVDF exhibits the same FT-IR characteristic peaks as the original PVDF membrane: a C-H stretching vibration at 2983 cm^−1^, a CH_2_ bending vibration at 1405 cm^−1^, a C-C vibration at 1186 and 879 cm^−1^ and a CF_2_ stretching vibration at 840 cm^−1^. After alternating the TTN assembly of d-UiO, TUT-UiO-4-TTN@PVDF has a C=O vibration at 1654 cm^−1^ of d-UiO-4 and UiO-66-NH_2_ and asymmetric tensile vibration at 768 and 661 cm^−1^ of Zr-O. The successful loading of d-UiO-4 on the PVDF membranes was again demonstrated [39]. However, in Figure 4b,c, d-UiO-4 shows weak crystal and low peak intensity in the crystal spectra of d-UiO-4@TUT-PVDF, which is attributed to the secondary etching and coordination of d-UiO by TA. Moreover, a double TTA complex continued to react and coordinate with the amino group and metal cluster on the surface of d-UiO, reducing the original ecological load of d-UiO on the PVDF membrane surface.

To verify the above results, TUT-UiO-4-TTN@PVDF was fully scanned by XPS; as can be seen from Figure 5a, TUT-UiO-4-TTN@PVDF shows strong peaks of F1s (688 eV), O1s (530 eV) and N1s (399 eV), which are attributed to the C-F bond of the PVDF, the O element introduced by d-UiO, TTN and PVDF, the N-H bond of the amino group on the surface of d-UiO and the N element in the TTA complex. In addition, the C1s in Figure 5b have four peaks, which are 284.5 eV C-C and C-H, 286 eV C-O, O-C=O for 288 eV and C-F for 289 eV, respectively. Moreover, 485 eV and 464 eV in Ti2p correspond to peaks of Ti 2p3/2 and Ti 2p1/2, respectively (Figure 5g), indicating the successful complexation of Ti^4+^ with TTA. The 182 eV and 184.6 eV in Zr3d correspond to the peaks of Zr 3d5/2 and Zr 3d3/2, respectively, representing the successful introduction of Zr^4+^, but the peak value is smaller, which also verifies the etching and coordination of the TA and TTA complex on d-UiO.

### 3.3. Performance of TUT-UiO-TTN@PVDF Membranes

The original PVDF membrane exhibited hydrophobic properties, as shown in Figure 6a. The d-UiO was loaded onto the membrane as a hierarchical structure and, with its inherent hydrophilicity, further enhanced the hydrophilic performance of the membrane. Therefore, the TUT-UiO-TTN@PVDF membrane showed that the contact angle was reduced 10° compared with the PVDF membrane. The hydrophilicity of TUT-UiO-TTN@PVDF does not change significantly with the further acetic acid modification of d-UiO, which is the main factor involved in the morphology and surface properties of the TUT layer. The TUT covering UiO nanocrystals produces a unique hydrophilic layered structure. As a result, the TUT mixed layer developed on the membrane surface facilitates superior water wettability.

Figure 6b–e shows the mechanical strength of the TUT-UiO-TTN@PVDF membrane compared with the original PVDF membrane. After modification by the TUT layer, the elongation at the break of the TUT-UiO-4-TTN@PVDF membrane is 24% higher than that of the original PVDF membrane. Similarly, the elongation at the break of the TUT-UiO-4-TTN@SPVDF and TUT-UiO-4-TTN@PP membrane are higher than that of the original SPVDF and PP membrane, with 3% and 22%, respectively, indicating that the binding force between the TUT layer and the base membrane is weaker than the molecular chains. This increases the possibility of interface separation during the stretching of the modified membrane while not affecting the mechanical strength of the PVDF substrate, thus improving the mechanical properties.

A thermogravimetric analysis (TGA) was performed in the air atmosphere at a rate of 5 °C·min^−1^, as shown in Figure 7a–c. The weightlessness process is mainly divided into three steps: (1) weightlessness before 100 °C involves the evaporation of water and the solvent; (2) weight loss at 100–350 °C involves the decomposition of TTA, the removal of monocarboxylic acid ligand and the dihydroxylation of Zr_6_ clusters; (3) and the weight loss at 350–500 °C involves the decomposition and oxidation of the PVDF membrane; (4) additionally, after 500 °C, the decomposition of the BDC-NH_2_ and the formation of inorganic ZrO_2_ residue begins. The weight loss rate of the TUT-UiO-TTN@PVDF membrane was higher than that of PVDF polymer (69.2%) and PVP-Zr-MOF-F (70.9%), indicating that PVP-Zr-MOF-F was successfully loaded into the membrane.

### 3.4. Comprehensive Evaluation of Gas Separation Performances

#### 3.4.1. Influence of Defect Strategy on Gas Separation Performance

d-UiO combines nanoscale, abundant defects and a large number of micropores, showing great potential in regard to gas adsorption and separation. The CO_2_, CH_4_ and N_2_ adsorption performance of d-UiO-3 are shown in Figure 8a. At 273 K and 100 kPa, the equilibrium adsorption capacities of CO_2_, CH_4_ and N_2_ of d-UiO-3 are 55.6 cm^3^·g^−1^, 11.3 cm^3^·g^−1^ and 2.5 cm^3^·g^−1^, respectively, which indicates the -NH_2_ group of d-UiO-3 ligand has a strong adsorption capacity for CO_2_ molecules. More importantly, the adsorption capacity of d-UiO-3 for CO_2_ is significantly higher than that of CH_4_ and N_2_ at the same temperature and pressure, which means that d-UiO has the ability to selectively CO_2_ from CH_4_ and N_2_. Furthermore, the permeability and selectivity of d-UiO-3@TUT-PVDF membranes for H_2_, CO_2_, CH_4_ and N_2_ are shown in Figure 8b. With the support of the large pores of the PVDF membrane, the rich amino groups and the strong complexation of the TUT layer, CO_2_ molecules showed a strong permeability and selectivity for CO_2_/N_2_ and CO_2_/CH_4_, with 27 and 18, respectively.

Figure 8c,d shows the permeability and selectivity of TUT-PVDF membranes with different defect degrees. It can be seen that, compared with the original PVDF membrane, the H_2_, N_2_ and CH_4_ permeation fluxes of the membrane containing the TUT layer are reduced, which is due to the increasing of the dense selective layer that reduces the permeability of non-adsorbed molecules. The permeable flux of CO_2_ increases when defects in the interlayer material are decreased. When d-UiO-3 is selected as the interlayer, the permeable flux of the membrane reaches the optimal level, indicating that an appropriate amount of acetic acid can promote the removal of ligands and the coordination unsaturation of some metal sites, providing more adsorption active sites for the gas and retaining amino functional groups without affected the adsorption of CO_2_. With the further increase in d-UIO defects, the crystal quality, amino ligand and the complexation of the TTN layer decrease, and the particle size of the complexation layer and the non-selective macropores also increase. In addition, the amounts of CO_2_/N_2_ and CO_2_/CH_4_ in the d-UiO-3@TUT-PVDF membrane are 2164% and 1388% higher than original PVDF membrane. Due to the strong interactions between d-UiO and TTA complexes containing defects, the phenol hydroxyl groups on the surface of the TTA complex are bound to d-UiO by hydrogen bonding and van der Waals forces, forming a dense structure on the interlayer surface separation layer. Therefore, the TUT-PVDF membrane exhibits a better gas selectivity performance than the original PVDF membrane (Table 2).

#### 3.4.2. Influence of Gas Infiltration Activation Energy

In order to study the effects of temperature on the properties of CO_2_, N_2_ and CH_4_, as well as the selectivity of CO_2_/N_2_ and CO_2_/CH_4_, permeability tests were carried out on TUT-UiO-3-TTN@PVDF at different temperatures under the same pressure, as shown in Figure 9. It can be seen that the permeability of CO_2_, N_2_ and CH_4_ increases linearly with the increase in temperature, while the selectivity of CO_2_/N_2_ and CO_2_/CH_4_ decreases gradually, which is mainly due to the fact that the kinetic energy of the gas with temperature increases, thereby accelerating the diffusion performance. At the same time, a higher temperature enhances the activity of the PVDF molecular chain, reduces the diffusion resistance of the gas through the membrane, and thus improves the permeability. However, the increase in temperature is not conducive to the dissolution of CO_2_ in the membrane and has little effect on the dissolution of N_2_, which ultimately leads to a decrease in selectivity. The osmotic activation energies of CO_2_, N_2_ and CH_4_ (Ea, kJ·mol^−1^) are obtained by linearizing the Arrhenius formula. The results show that the Ea of CO_2_, N_2_ and CH_4_ are 0.55 kJ·mol^−1^, 9.56 kJ·mol^−1^ and 7.48 kJ·mol^−1^, respectively, indicating that the permeation behavior of the TUT-UiO-3-TTN@PVDF membrane to different gasses is temperature dependent with the order of N_2_ > CH_4_ > CO_2_.

Gas permeability is affected by the molecular diameter, and the larger the molecular diameter, the smaller the amount of permeability; additionally, the dependence on temperature also increases. However, the combined effects of competitive adsorption and competitive diffusion make the Ea of CO_2_ and CH_4_ lower. The long-term stability of TUT-UiO-TTN@PVDF was further studied at 273 K and 100 kPa with a sample membrane area of 4.9 cm^2^ (Figure 9f). After 40 h continuous operation, both CO_2_ transmittance and CO_2_/N_2_ selectivity remained unchanged, indicating excellent operational stability. The TUT layer is more stable because of the strong interaction between the defective MOFs and the supporting layer. Compared to the original PVDF membrane, the TUT-UiO-4-TTN@PVDF membrane showed less interlayer material shedding during the same time, which is why the interlayer material d-UiO-4 of TUT-UiO-4-TTN@PVDF membrane has a significant amount of incompletely coordinated Zr^4+^. In addition to amino groups, they interact more strongly with the TA support layer.

#### 3.4.3. Study on Separation Performance of Gas Mixture

The selectivity of TUT-UiO-TTN@PVDF was investigated with the mixture ratio of typical flue gas (CO_2_/N_2_, 15/85, *v*/*v*) and raw natural gas (CO_2_/CH_4_, 10/90, *v*/*v*), as shown in Figure 10a. The permeability of TUT-UiO-3-TTN@PVDF in the simulated flue gas and natural gas is 11,567 GPU and 10,681 GPU, respectively, and the CO_2_/N_2_ and CO_2_/CH_4_ selectivity are 24 and 29, respectively, indicating that CO_2_ has competitive diffusion with N_2_ and CH_4_ during the gas mixture penetration, which is mainly due to the presence of a large number of CO_2_-philic defect sites and amino active functional groups in the TUT layer. Among these, the CO_2_-philic defect sites in d-UiO formed interlayer screening channels in the membrane, which enhanced the selective separation of CO_2_. The amino active functional groups in the TUT layer construct a channel to promote CO_2_ transfer and further improve the permeability of CO_2_.

The distribution and adsorption energy of flue gas and raw gas in d-UiO were analyzed using simple MC simulations. At first, 15 CO_2_ molecules and 85 N_2_ or 10 CO_2_ molecules and 90 CH_4_ molecules are placed in the cage to track the chemical composition of the flue gas and raw natural gas. The results show that CO_2_ molecules have strong polarity and high quadrupole moments and that they occupy the high energy adsorption sites of d-UiO for flue gas and natural gas, respectively, while the non-polar N_2_ molecules and the larger CH_4_ molecules are distributed at the low energy sites or residual sites. In addition, the density of CO_2_ molecules is higher in defect regions because these defects introduce additional high-energy sites. The N_2_ molecule is less affected by defects due to its low adsorption energy, so the change is not as obvious as that for CO_2_.

The interaction diagram of the mixture with the MOFs shows that there are transient dipole-induced dipole interactions or dipole–dipole interactions between the gas molecules of the two systems and the MOF due to the attraction and interaction between the CO_2_ molecules and the MOFs. In addition, according to the curve of adsorption energy in the mixture shown in Figure 10i, the adsorption energy of CO_2_ is much higher than that of N_2_ and CH_4_, which is due to the polarity of CO_2_ and the strong interaction with the adsorption sites, which proved that d-UIO has high selectivity in regard to CO_2_ in the separation system of CO_2_/N_2_ and CO_2_/CH_4_ and is consistent with the experimental results.

## 4. Conclusions

In this work, the strategies of step-by-step assembly, alternate complexation and cooperative coordination are adopted to successfully construct TUT-UiO-TTN@PVDF with a high performance. The defective d-UiO nanocrystals not only enhance the adsorption of CO_2_ but also accelerate the diffusion of CO_2_ through the defect cavities, which significantly improves the separation performance as well as provides a faster CO_2_ transmission channel, resulting in a 56% increase in *S*_BET_ and micropores. The hydrophilicity of TUT-UiO-TTN@PVDF membrane is enhanced compared with the original PVDF membrane and the contact angle is reduced by 10°. The TUT-UiO-3-TTN@PVDF membrane achieves a CO_2_ permeation flux of 14,294 GPU and selectivities for CO_2_/N_2_ and CO_2_/CH_4_ of 27 and 18, respectively. The TUT-UiO-TTN@PVDF membrane has great potential for selective CO₂ separation, making it suitable for flue gas treatment and natural gas purification. This work of improving membrane separation performance through material design and structural regulation provides important technical support for the field of energy and environment in the era of carbon reduction. Meanwhile, gas separation membrane technology has broad prospects in the fields of energy saving, environmental protection and industrial gas purification, and it is necessary to continue to examine the performance of membrane materials in the future while paying attention to the feasibility and economy of their industrial applications to promote the development of gas separation membranes.

## Figures and Tables

**Figure 1 nanomaterials-15-00279-f001:**
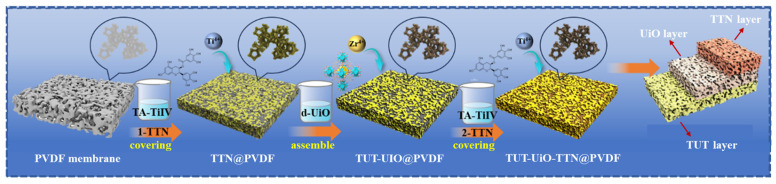
A schematic of the TUT-UiO-TTN@PVDF membrane was prepared by the strategies of step assembly, alternate complexation and cooperative coordination.

**Figure 2 nanomaterials-15-00279-f002:**
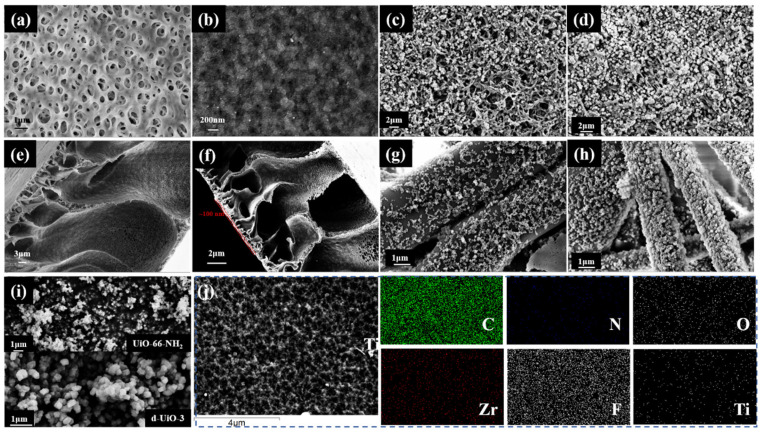
SEM images of the (**a**) surface and (**e**) section of PVDF, (**b**,**f**) of TUT-UiO-TTN@PVDF, (**c**,**g**) of TUT-UiO-TTN@SPVDF, (**d**,**h**) TUT-UiO-TTN@PP, (**i**) UiO-66-NH_2_ and d-UiO-3; (**j**) an EDS analysis of the TUT-UiO-3-TTN@PVDF samples.

**Figure 3 nanomaterials-15-00279-f003:**
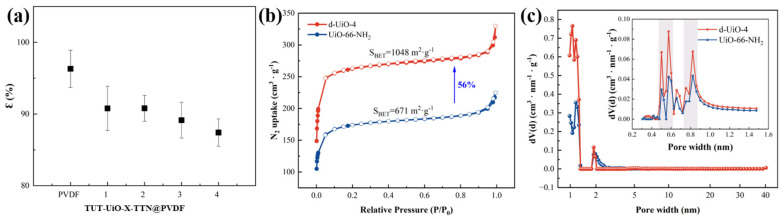
(**a**) Porosity of TUT−UiO−TTN@PVDF, (**b**) N_2_ adsorption−desorption isotherm, and (**c**) pore size distribution.

**Figure 4 nanomaterials-15-00279-f004:**
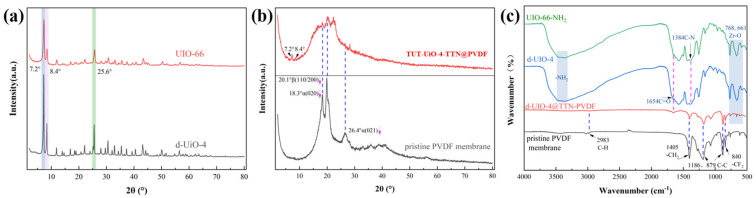
XRD images of (**a**) d−UiO−4 and UiO−66 and (**b**) d−UiO−4@TUT−PVDF; (**c**) FT−IR images of d−UiO−4, UiO−66−NH_2_ and TUT−UiO−4−TTN@PVDF.

**Figure 5 nanomaterials-15-00279-f005:**
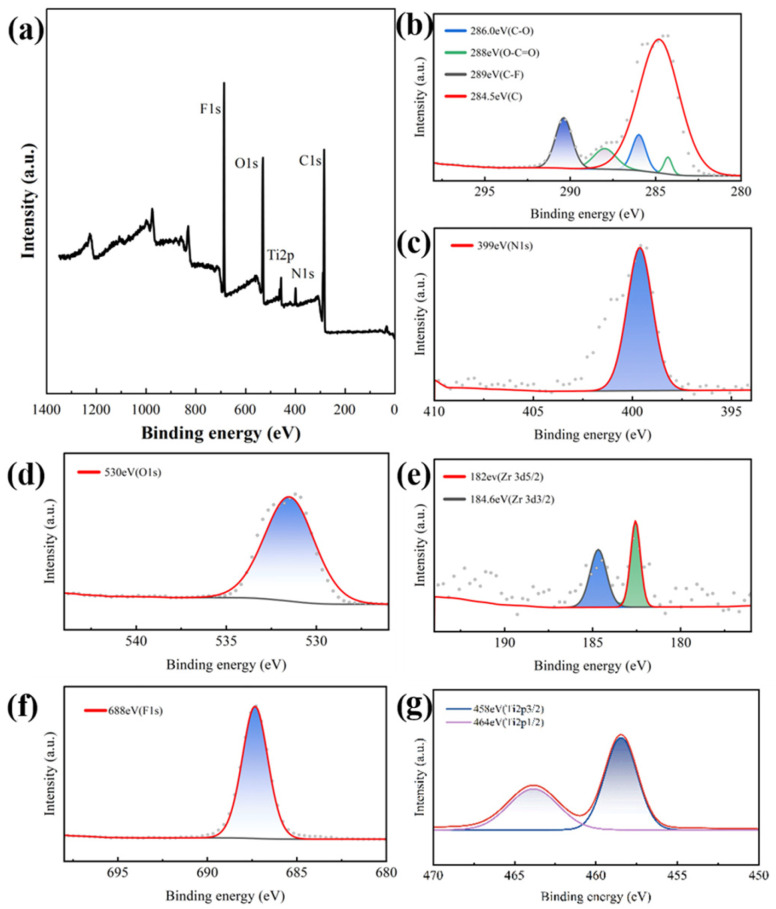
XPS of (**a**) the TUT-UiO-4-TTN@PVDF membrane for (**b**) C1s, (**c**) N1s, (**d**) O1s, (**e**) Zr3d, (**f**) F1s and (**g**) Ti2p.

**Figure 6 nanomaterials-15-00279-f006:**
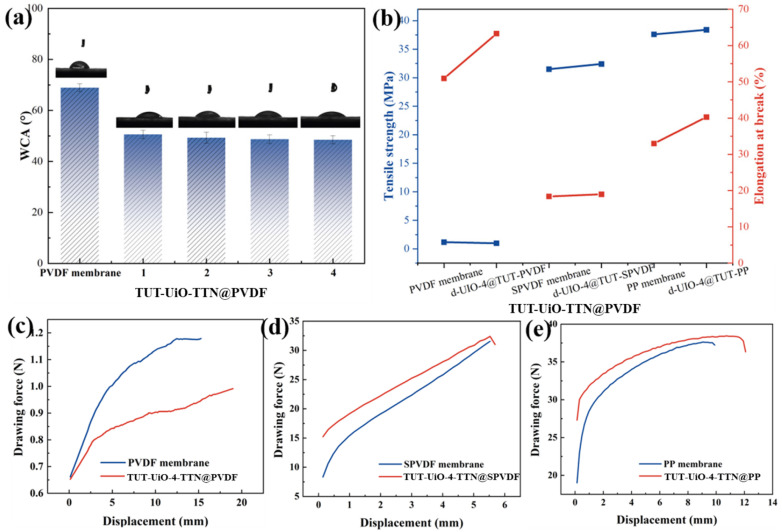
Contact angle of the (**a**) TUT-UiO-TTN@PVDF, (**b**) tensile strength and elongation, mechanical strength of (**c**) TUT-UiO-4-TTN@PVDF membrane, (**d**) TUT-UiO-4-TTN@SPVDF and (**e**) TUT-UiO-4-TTN@PPwith original membrane.

**Figure 7 nanomaterials-15-00279-f007:**
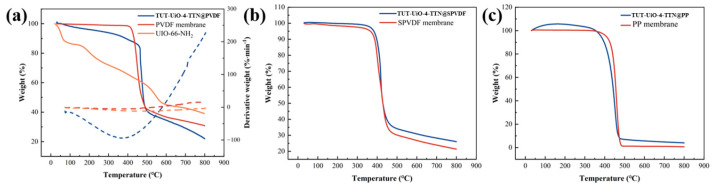
TGA images of (**a**) a TUT-UiO-4-TTN@PVDF membrane, (**b**) a TUT-UiO-4-TTN@SPVDF membrane and (**c**) a TUT-UiO-4-TTN@PP membrane.

**Figure 8 nanomaterials-15-00279-f008:**
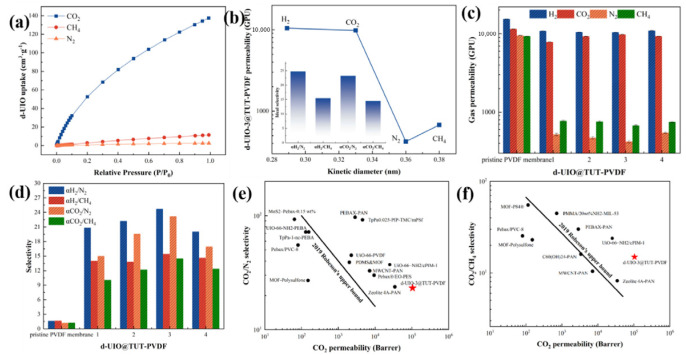
(**a**) The adsorption curves of CO_2_, CH_4_ and N_2_ of d-UiO-3; (**b**) the permeation flux and selectivity of TUT-UiO-3-TTN@PVDF for H_2_, CO_2_, CH_4_ and N_2_; (**c**) the permeability and (**d**) selectivity of TUT-PVDF with different defect degrees; comparison of the (**e**) CO_2_/N_2_ and (**f**) CO_2_/CH_4_ selectivity of TUT-UiO-4-TTN@PVDF with other MOF-containing MMMs.

**Figure 9 nanomaterials-15-00279-f009:**
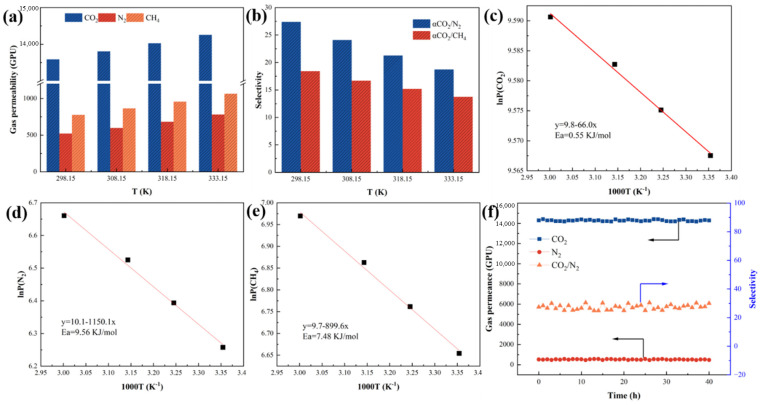
(**a**) The permeability of the TUT-UiO-4-TTN@PVDF membrane at different temperatures; (**b**) the selectivity, osmotic activation energy of the membrane for (**c**) CO_2_, (**d**) N_2_ and (**e**) CH_4_; and (**f**) the long-term stability of the TUT-UiO-TTN@PVDF membrane.

**Figure 10 nanomaterials-15-00279-f010:**
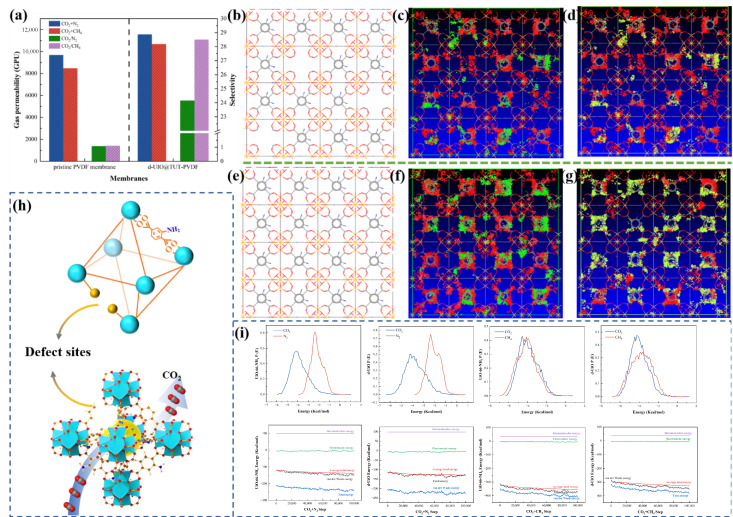
(**a**) The permeability and selectivity of the TUT-UiO-4-TTN@PVDF membrane for the mixture, (**b**) d-UiO crystal model, (**c**) d-UiO adsorption simulation of CO_2_/N_2_, (**d**) d-UiO adsorption simulation of CO_2_/CH_4_, (**e**) UiO-66-NH_2_ crystal model, (**f**) adsorption simulation of CO_2_/N_2_ by UiO-66-NH_2_, (**g**) adsorption simulation of CO_2_/CH_4_ by UiO-66-NH_2_, (**h**) schematic diagram of selective gas adsorption by d-UiO, (**i**) interaction energy of gas adsorption by d-UiO and UiO-66-NH_2_.

**Table 1 nanomaterials-15-00279-t001:** Pore structure of UiO-66-NH_2_ and XA-UiO-66-NH_2_.

Samples	*S*_BET_ ^a^ (m^2^·g^−1^)	*S*_Micro_ ^b^ (m^2^·g^−1^)	*V*_Micro_ ^c^ (cm^3^·g^−1^)	*V*_Total_ ^d^ (cm^3^·g^−1^)	Pore Size (nm)
d-UiO-4	1048	1013	0.40	0.51	1.95
UiO-66-NH_2_	671	616	0.25	0.35	1.17

^a^: Brunauer–Emmett–Teller (BET) specific surface area. ^b^: Surface area of micropores. ^c^: Microporous volume. ^d^: Total pore volume.

**Table 2 nanomaterials-15-00279-t002:** Comparison of CO_2_/N_2_ selectivity of TUT-UiO-4-TTN@PVDF membranes with other MOF-containing MMMs.

Samples	Condition	Permeability	Selectivity		Ref.
			CO_2_/N_2_	CO_2_/CH_4_	
UiO-66−NH_2_/cPIM-1	-	2504 GPU	37.20	23.80	[28]
TpPa0.025-PIP-TMC/mPSf	298 K, 0.15 MPa	854 GPU 128 bar	148.00	-	[40]
TpPa0.025-PIP-TMC/mPSf	0.5 MPa	456 GPU	92.00	-	[40]
PDMS&MOF	35 °C, 1.0 bar	1990 GPU	39.00	-	[41]
TpPa-1-nc-PEBA	298 K, 0.3 MPa	156.9 bar	72.00	-	[42]
UiO-66-NH_2_-PEBA	298 K, 0.2 MPa	130 bar	72.00	-	[29]
MoS2–Pebax-0.15 wt%	30 °C, 2 bar	64 bar	93.00	-	[43]
Pebax^®^/EO-PES	57 °C, 1.5 psig	940 GPU	30.00	-	[44]
Pebax/PVC-8	-	8 GPU	55.30	25.30	[45]
MOF-Polysulfone	-	15 GPU	27.00	23.00	[46]
Zeolite 4A-PAN	-	3457 GPU	23.80	8.18	[47]
UiO-66-PVDF	-	225 GPU	45.00	-	[30]
C60(OH)24-PAN	-	338 GPU	40–55	15–17	[48]
MWCNT-PAN	2 bar, 27 °C	7090 bar	32.90	10.40	[49]
PEBAX-PAN	-	287 GPU	97.00	30.00	[50]
TUT-UiO-3-TTN@PVDF	30 °C, 1.5 bar	10,439 GPU	23.20	14.90	This work

## Data Availability

The original contributions presented in the study are included in the article, further inquiries can be directed to the corresponding author.

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
