# Peer review of "Self-Assembled Sandwich-like Mixed Matrix Membrane of Defective Zr-MOF for Efficient Gas Separation"

_nanomaterials, 2025, doi:10.3390/nano15040279_

Round 1

Reviewer 1 Report

Comments and Suggestions for Authors

The manuscript presents a strategy for preparing high-performance mixed matrix membranes (MMMs) for gas separation by incorporating Zr-MOF via a step-by-step assembly process. While the authors propose a promising approach for enhancing membrane performance, supported by experimental data, several critical issues need to be addressed to improve the manuscript's quality. Firstly, the explanations provided lack depth and fail to include essential details necessary for a comprehensive understanding of the work. Additionally, certain sections are unclear, making it challenging for readers to follow the reasoning and methodology. The manuscript requires substantial improvements in clarity and coherence to enhance readability. Moreover, the organization of the paper is inadequate, and the descriptions of the experimental procedures lack sufficient detail, making it difficult to assess the reproducibility of the study. Given these significant concerns regarding clarity, organization, and completeness, I do not find the manuscript suitable for publication in Nanomaterials in its current form.

1.      The manuscript should be checked carefully to improve the English. There are some spelling mistakes and unnecessary wordiness are present throughout the writing and need to be addressed.

In the introduction:

2.      Line 38: The phrase may be simplified to “… to minimize mass transfer resistance…” for conciseness.

3.      Terminology Consistency: Ensure consistent terminology throughout the manuscript, for example, MOF-based mixed matrix membrane (Line 42) vs. MOF mixed matrix membrane (Line 20). Standardizing the terminology will improve clarity. 

4.      Line 52: Replace “interweave” with “integrate”, as "integrate" more accurately conveys the interaction between the components in this context.

5.      Line 56: The permeability unit needs revision, as “bar” is not the correct unit for permeability. Please use the appropriate unit.

6.      Lines 57–64: The description in this section is not precise and may mislead readers into thinking the focus is on Pebax-based membranes or surface functionalization rather than the intended topic. Please rephrase to improve clarity and ensure the main point is accurately conveyed.

7.      Line 65: The term "biomatrix phenolic network" is unclear, as the manuscript does not mention any bio-based materials. Please clarify or revise.

8.      Line 66: Suggest revising "step assembly" to "step-by-step assembly" for consistency and accuracy.

In Materials and Methods

9.      Line 76, The abbreviation TA for tannic acid is missing.

10.  PVDF Identification: Clarify whether PVDF refers to the polymer precursor or a PVDF MF membrane. If it refers to the polymer, please specify its molecular weight (MW). If it refers to the membrane, provide details on its pore size.

11.  Section 2 lacks a description of the gas permeation testing methodology. Please include the relevant details to ensure completeness and reproducibility.

12.  Section 2.3: (a) Line 95: Since “acetic acid” is used for the synthesis of defective UiO-66-NHâ‚‚, please list it in Section 2.1 Materials for completeness. (b) Line 98: Clarify what is meant by "the number of samples"—does this refer to?

13.  Section 2.4: This section requires rephrasing to improve clarity and remove redundant information. Additionally, essential details are missing, making it difficult to follow the methodology. Please address the following points:
(a) Line 100: As mentioned in Q8, is "metal-phenolic" referring to the same material as previously discussed? If so, please ensure consistent terminology.
(b) Line 101: Does "microfiltration membrane" refer to the PVDF membrane? If so, please use consistent terminology throughout the manuscript to enhance readability.
(c) Line 102: The description of the washing and storage process is unclear. Please specify:

·   How many times is the membrane washed with methanol?

·   Where is it stored after washing?

·   Why is at least 12 hours required for storage?
(d) Related to Q10: If the PVDF MF membrane is self-prepared, please provide an appropriate reference for its preparation method.
(e) Specify the concentrations of TA and Ti-BALDH used in the process.
(f) Figure 1 is not referenced in the manuscript. Please either cite it appropriately or remove it if unnecessary.

14.  Figure 1: While the figure presents detailed material structures, it lacks clear labels and step descriptions, making it difficult to interpret. Please address the following concerns:

·  The top row does not specify what molecular lattices are being depicted—please provide labels.

·  Does the step-by-step assembly membrane have distinct names for each stage? If so, please include them for clarity.

·  The top-right elements in each membrane diagram are unclear—what do they represent?

·  The overall figure size is too small, making it difficult to understand the content. Consider enlarging or restructuring it for better readability.

·  In the resulted membrane, what do the three layers represent? Why do they all have the same pore structure but only differ in color? Please clarify.

·  The figure introduces a new term (e.g., TA-TIV) that is not explained in the text. Please define it.

15.  Lines 127–128: The term "micropore SBET" should be revised to show the full name in the first for clarity and correctness.

In Section 3: Results and discussion

16.  Section 3.1:

(a) Lines 132–144: The description does not effectively correlate with Figure 2. To improve clarity, please cite specific figure where appropriate to ensure a stronger connection between the text and the images.

(b) Figure 2f: The claim that the membrane thickness is 100–300 nm based solely on SEM images is insufficient. Consider further verification using EDS cross-section line mapping to analyze the average thickness distribution more accurately.

(c) Line 133: The term "smooth" should be removed, as the images do not provide a quantitative measure of surface smoothness. Additionally:

·     Figures 2a and 2b have different scales, making it difficult to effectively compare surface structures.

·     If roughness is a key discussion point, consider using AFM for a more precise analysis.

(d) Lines 134–135: The claim that UiO is uniformly dispersed on the membrane surface requires further clarification. Since d-UiO is stated to be 20–50 nm, the resolution of SEM-EDS may not be sufficient to determine uniform dispersion or detect potential agglomeration. Please provide additional supporting evidence or discussion.

(e) Figure 2j: What specific sample does this represent? Please clarify in the figure caption and text.

(f) Lines 139–140: How was it determined that the defective active sites of d-UiO increased and that more d-UiO was involved in the formation of the TUT functional layer? Please provide experimental evidence or a clear explanation to support this claim.

17.  Lines 145–150: For the discussion on the defect modification of UiO-66, please cite relevant literature to support this claim and strengthen the scientific basis of your argument.

18.  There are instances where d-UiO-3 is used and others where d-UiO-4 is mentioned. Are these two forms of d-UiO identical, or is there a distinction between them? Can these materials be directly compared in the same context? Additionally, could you clarify which specific form of d-UiO@TUT-PVDF is used in the experiments?

19.  Figure 3: The panels of Figure 3 should be reordered to follow the logical sequence discussed in the manuscript. For example, Figure 3a should be placed at the end of the figure legend to align with the narrative flow.

20.  Line 184: Is there any supporting literature or experimental evidence for surface etching of the PVDF membrane by TA? Please provide references or data to back this statement.

21.  PP and SPVDF in the Study: The use of PP and SPVDF is mentioned, but the rationale for including these materials is not explained. Furthermore, there is no mention of their individual gas separation performance, which suggests that their importance in the study may be limited. Consider placing this information in the supplementary materials if it is not critical to the main discussion.

22.  Long-term Stability for CO2/N2 Separation: Please provide more details on the operating conditions for the long-term stability tests, such as temperature, pressure, and the type of system used. These details are essential for understanding the experimental setup and the relevance of the results.

Comments on the Quality of English Language

The manuscript should be checked carefully to improve the English. There are some spelling mistakes and unnecessary wordiness are present throughout the writing and need to be addressed.

Author Response

Reviewer 1:

Comments 1: The manuscript should be checked carefully to improve the English. There are some spelling mistakes and unnecessary wordiness are present throughout the writing and need to be addressed.

Response 1: Thanks for your constructive comments. We have corrected them step by step.

In the introduction:

Comments 2: Line 38: The phrase may be simplified to “… to minimize mass transfer resistance…” for conciseness.

Response 2: Thank you for pointing this out. We have changed the phrase to make it simplified.

Comments 3: Terminology Consistency: Ensure consistent terminology throughout the manuscript, for example, MOF-based mixed matrix membrane (Line 42) vs. MOF mixed matrix membrane (Line 20). Standardizing the terminology will improve clarity.

Response 3: Thank you for pointing this out. We have corrected them, which all named as MOF-based mixed matrix membrane.

Comments 4: Line 52: Replace “interweave” with “integrate”, as "integrate" more accurately conveys the interaction between the components in this context.

Response 4: Thank you for pointing this out. We have replaced “interweave” with “integrate”.

Comments 5: Line 56: The permeability unit needs revision, as “bar” is not the correct unit for permeability. Please use the appropriate unit.

Response 5: Thank you for pointing this out. We have corrected the unit as “Barrer”, which it is a unit used to assess the permeability of gases. ‌Barrer is a unit used to assess gas permeability, defined as the permeability of a gas through a polymer film within a given unit thickness (cm).

Comments 6: Lines 57–64: The description in this section is not precise and may mislead readers into thinking the focus is on Pebax-based membranes or surface functionalization rather than the intended topic. Please rephrase to improve clarity and ensure the main point is accurately conveyed.

Response 6: Thank you for pointing this out. We have changed this section to make sure they convey the right meaning and fit in with the topic.

Comments 7: Line 65: The term "biomatrix phenolic network" is unclear, as the manuscript does not mention any bio-based materials. Please clarify or revise.

Response 7: Thank you for pointing this out. We have corrected the sentences make it more accurately. For example: In this work, the membrane coated with metal-phenolic was constructed by stepping assembly, alternating complexation and cooperative coordination.

Comments 8: Line 66: Suggest revising "step assembly" to "step-by-step assembly" for consistency and accuracy.

Response 8: Thank you for pointing this out. We have corrected "step assembly" to "step-by-step assembly" for consistency and accuracy.

In Materials and Methods

Comments 9: Line 76, The abbreviation TA for tannic acid is missing.

Response 9: Thank you for pointing this out. We have added the TA for the abbreviation of tannic acid.

Comments 10: PVDF Identification: Clarify whether PVDF refers to the polymer precursor or a PVDF MF membrane. If it refers to the polymer, please specify its molecular weight (MW). If it refers to the membrane, provide details on its pore size.

Response 10: Thank you for pointing this out. PVDF refers to the polymer precursor with molecular weight of 950 kDa.

Comments 11: Section 2 lacks a description of the gas permeation testing methodology. Please include the relevant details to ensure completeness and reproducibility.

Response 11: Thank you for pointing this out. We have added the description of gas penetration test methods to make it complete and accurate.

Comments 12: Section 2.3: (a) Line 95: Since “acetic acid” is used for the synthesis of defective UiO-66-NHâ‚‚, please list it in Section 2.1 Materials for completeness. (b) Line 98: Clarify what is meant by "the number of samples"--does this refer to?

Response 12: Thank you for pointing this out. We have list “Acetic acid” in Section 2.1. Besides, “the number of samples” refer to acetic acid was added to the mixed solution at a fixed molar ratio (Zr: NH2-BDC: DMF: acetic acid = 1:1:500:15, 30, 70, 100). d-UiO-1, d-UiO-2, d-UiO-3 and d-UiO-4 refer to use different molar ratio of acetic acid.

Comments 13: Section 2.4: This section requires rephrasing to improve clarity and remove redundant information. Additionally, essential details are missing, making it difficult to follow the methodology. Please address the following points:

(a) Line 100: As mentioned in Q8, is "metal-phenolic" referring to the same material as previously discussed? If so, please ensure consistent terminology.

(b) Line 101: Does "microfiltration membrane" refer to the PVDF membrane? If so, please use consistent terminology throughout the manuscript to enhance readability.

(c) Line 102: The description of the washing and storage process is unclear. Please specify: How many times is the membrane washed with methanol?

Where is it stored after washing?

Why is at least 12 hours required for storage?

(d) Related to Q10: If the PVDF MF membrane is self-prepared, please provide an appropriate reference for its preparation method.

(e) Specify the concentrations of TA and Ti-BALDH used in the process.

(f) Figure 1 is not referenced in the manuscript. Please either cite it appropriately or remove it if unnecessary.

Response 13: Thank you for pointing this out. Here are the answers to the questions:

(a) “metal-phenolic” referring to the same material as previously discussed, we have replaced the consistent terminology.

(b) "microfiltration membrane" refer to the PVDF membrane, we have corrected to use consistent terminology throughout the manuscript.

(c) The membrane is washed twice with methanol and placed on a flat plate in the laboratory. The storage time of 12 hours is need to remove surface impurities of membrane.

(d) PVDF refers to the polymer precursor.

(e) We used 5% TA and 10% Ti-BALDH in the process.

(f) Figure 1 is referenced in the manuscript (Line 114).

Comments 14: Figure 1: While the figure presents detailed material structures, it lacks clear labels and step descriptions, making it difficult to interpret. Please address the following concerns:

The top row does not specify what molecular lattices are being depicted—please provide labels.

Does the step-by-step assembly membrane have distinct names for each stage? If so, please include them for clarity.

The top-right elements in each membrane diagram are unclear—what do they represent?

The overall figure size is too small, making it difficult to understand the content. Consider enlarging or restructuring it for better readability.

In the resulted membrane, what do the three layers represent? Why do they all have the same pore structure but only differ in color? Please clarify.

The figure introduces a new term (e.g., TA-TIV) that is not explained in the text. Please define it.

Response 14: Thank you for pointing this out.

Figure 1 is a schematic of the synthesis process. The name of each stage has been shown in the synthesis process, the top-right elements in each membrane diagram represent a more specific membrane structure.

Besides, we've enlarged the picture to make it easier to understand.

The schematic drawing of three layers represents different materials, the bottom layer is TTN@PVDF, the middle is TUT-UIO@PVDF, the top is TUT-UIO-TTN@PVDF. TA-TiIV is TA methanol solution and Ti-BALDH methanol solution, we have explained it in the text.

Comments 15: Lines 127–128: The term "micropore SBET" should be revised to show the full name in the first for clarity and correctness.

Response 15: Thank you for pointing this out. The specific surface area (SBET)was calculated using the Brunauer-Emmett-Teller (BET) equation (Line 143).

In Section 3: Results and discussion

Comments 16: Section 3.1:

(a) Lines 132–144: The description does not effectively correlate with Figure 2. To improve clarity, please cite specific figure where appropriate to ensure a stronger connection between the text and the images.

(b) Figure 2f: The claim that the membrane thickness is 100–300 nm based solely on SEM images is insufficient. Consider further verification using EDS cross-section line mapping to analyze the average thickness distribution more accurately.

(c) Line 133: The term "smooth" should be removed, as the images do not provide a quantitative measure of surface smoothness. Additionally: Figures 2a and 2b have different scales, making it difficult to effectively compare surface structures. If roughness is a key discussion point, consider using AFM for a more precise analysis.

(d) Lines 134–135: The claim that UiO is uniformly dispersed on the membrane surface requires further clarification. Since d-UiO is stated to be 20–50 nm, the resolution of SEM-EDS may not be sufficient to determine uniform dispersion or detect potential agglomeration. Please provide additional supporting evidence or discussion.

(e) Figure 2j: What specific sample does this represent? Please clarify in the figure caption and text.

(f) Lines 139–140: How was it determined that the defective active sites of d-UiO increased and that more d-UiO was involved in the formation of the TUT functional layer? Please provide experimental evidence or a clear explanation to support this claim.

Response 16: Thank you for pointing this out.

(a) we have added the cite of specific figure to ensure the connection between the text and the images.

(b) A line scan can show how the element content of a sample changes along a particular line direction. However, we conducted the evaluation through SEM of the membrane. This is very important and we will consider further verification using EDS cross-section line mapping to analyze the average thickness distribution more accurately.

(c) We have removed the term "smooth". Although roughness is not a key discussion point in this article, but this suggestion is valuable, and we will use it in future research.

(d) This is a very valuable suggestion. After SEM and EDS characterization, we believe that UiO is uniformly distributed, and we will consider more reasonable and accurate testing methods.

(e) Figure 2j represents d-UiO-3@TUT-PVDF.

(f) The original UiO-66-NH2 is a symbiotic aggregate of small grains. The morphology of d-UiO-3 after acetic acid regulation becomes more regular, the grain size increases, and the surface becomes more uniform and smoother, which is acetic acid acts as a regulator to accelerate crystal growth (Figure 2 i). In the unmodified membrane, d-UiO distribution in the membrane cavity is uneven and the load rate is low. After modification, the d-UiO is significantly increased, indicating that involved in the formation of the functional layer of TUT.

Comments 17: Lines 145–150: For the discussion on the defect modification of UiO-66, please cite relevant literature to support this claim and strengthen the scientific basis of your argument.

Response 17: Thank you for pointing this out. We have provided reference [27] to support this claim.

Comments 18: There are instances where d-UiO-3 is used and others where d-UiO-4 is mentioned. Are these two forms of d-UiO identical, or is there a distinction between them? Can these materials be directly compared in the same context? Additionally, could you clarify which specific form of d-UiO@TUT-PVDF is used in the experiments?

Response 18: The two forms of material are identical and can be compared under the same conditions. Besides, d-UiO-3@TUT-PVDF is used in the experiments.

Comments 19: Figure 3: The panels of Figure 3 should be reordered to follow the logical sequence discussed in the manuscript. For example, Figure 3a should be placed at the end of the figure legend to align with the narrative flow.

Response 19: Thank you for pointing this out. We have adjusted the content of this section to ensure that the content of the article corresponds to the order of the pictures.

Comments 20: Line 184: Is there any supporting literature or experimental evidence for surface etching of the PVDF membrane by TA? Please provide references or data to back this statement.

Response 20: Thank you for pointing this out. We have provided reference [28] to back this statement.

Comments 21: PP and SPVDF in the Study: The use of PP and SPVDF is mentioned, but the rationale for including these materials is not explained. Furthermore, there is no mention of their individual gas separation performance, which suggests that their importance in the study may be limited. Consider placing this information in the supplementary materials if it is not critical to the main discussion.

Response 21: Thank you for pointing this out. Mechanical properties are also an important indicator. With the introduction of PP and SPVDF, we want to show that the mechanical strength of d-UiO-3@TUT-PVDF is superior to that of PP and SPVDF.

Comments 22: Long-term Stability for CO2/N2 Separation: Please provide more details on the operating conditions for the long-term stability tests, such as temperature, pressure, and the type of system used. These details are essential for understanding the experimental setup and the relevance of the results.

Response 22: Thank you for pointing this out. The operating conditions were performed at 273 K and 100 kPa with a sample membrane area of 4.9 cm2 and continuous operation of 40 hours, which have been added in the revision manuscript (Line 341).

Reviewer 2 Report

Comments and Suggestions for Authors

Dear Authors,

I have carefully reviewed the manuscript “Self-assembled sandwich-like mixed matrix membrane of defective Zr-MOF for efficient gas separation.’’ The study presents interesting results and topic is highly relevant to the current in gas separation technologies. However, I have some critical concerns, and this study can be further improved for the scientific rigor, novelty and reproducibility based on my comments below. I believe these comments will help the author to improve their manuscript.

General Comments

·       Improve logical flow by ensuring smooth transitions from challenges to the motivation behind the study.

·       Clearly define the research gap by stating what is missing in current studies and how this work addresses those gaps.

·       Strengthen the introduction by explicitly outlining the expected contributions, particularly how the proposed membrane design outperforms previous designs.

·       Define key technical terms to make novel synthesis strategies accessible to non-specialist readers.

·       Provide purity details for all chemicals, especially critical reagents like ZrCl4, NHâ‚‚-BDC, and Ti-BALDH, as their quality can significantly impact synthesis results.

·       Ensure crucial experimental details for reproducibility, including stirring speed (rpm) during dissolution steps, cooling rate after solvothermal reactions, and centrifugation parameters.

·       Specify concentrations for TA, Ti-BALDH, and methanol solutions used in membrane modification.

·       Provide washing conditions to ensure reproducibility.

·       Maintain consistency in chemical symbols and formatting (e.g., NHâ‚‚-BDC instead of NH2-BDC).

·       Clearly indicate whether reagent ratios are in molar terms to avoid ambiguity.

·       Include more quantitative comparisons, such as tabulated data for particle sizes and defect density.

·       Strengthen the connection between structural/morphological changes and functional membrane performance, including permeability, antifouling behavior, and mechanical properties.

·       Revise unclear sentences for better clarity and readability, ensuring the manuscript is polished and professional.

Introduction

·       Line 31-33: "Petrochemical production, coal gasification………" the phrase "haze weather" is unclear. What does it mean?

·       Line 34-35: "Gas separation technology plays a key role ……" This is a broad statement. Specify which gas separation techniques are most effective or give examples and Mention how membrane separation compares to other techniques in terms of efficiency and cost.

·       Line 36-39: "Excellent membrane materials…….." Add an example of a material that meets these criteria to strengthen the argument.

·       Line 40-41: "Due to the high selectivity and large pore volume, MOF have……….” Please check the correct phrasing, it should be "MOFs have", not "MOF have". And "High selectivity" compared to what?

·       Line 45-46: "However, the limited interaction ……..” is there any study which tried to provide solution?

·       Line 50-64 (I think its relevant literature review?) The discussion on previous studies is informative but somewhat scattered. It would be helpful to structure this section better: First, discuss advances in polymer membranes. Then, introduce the role of MOFs in improving performance. Finally, highlight key challenges and gaps. Provide a brief summary of the key takeaways from these studies before introducing your research.

·       Line 65-72 (I hope its author`s Novelty of the Work): "In this work, a biomatrix phenolic network coated……...." This section should be more explicit about the novelty of the approach. How is this different from previous MOF-MMM approaches? Also, Define "step assembly, alternating complexation, and cooperative coordination".

Methodology

Materials Section (Lines 74-84):

·       Specify reagent purity, as this affects experimental outcomes. Additionally, gases (CHâ‚„, Nâ‚‚, COâ‚‚) should mention purity levels.

UiO-66-NHâ‚‚ Synthesis (Lines 85-91)

·       The dissolution and stirring step needs clarification regarding stirring speed and temperature.

·       The cooling process should specify whether it was natural cooling or controlled cooling.

·       The ethanol washing step should include the volume per washing cycle and the number of cycles.

Defective UiO-66-NHâ‚‚ Synthesis (Lines 92-98)

·       The phrase "stirred at room temperature for 10 min" should specify the stirring speed (e.g., 500 rpm).

·       It is unclear whether the reaction time and temperature were optimized or based on previous studies.

·       Question: Did you perform control experiments to confirm the defect formation? Consider adding references to confirm that the defect structure was obtained.

TFN Membrane Fabrication (Lines 99-109)

·       The concentration of TA and Ti-BALDH solutions should be mentioned.

·       "Sufficient adsorption of TA" is vague. Provide adsorption equilibrium time or reference data.

·       The term "stored for at least 12 h" should indicate storage conditions (temperature, humidity).

Characterization Section (Lines 113-128)

·       The SEM preparation method should specify the working voltage and magnification used for imaging.

·       The XRD analysis should specify scan range and step size.

·       TGA should indicate heating rate and atmosphere.

Results and Discussion

Clarity of SEM Image Descriptions (Lines 132-138)

·       The description of SEM images is somewhat qualitative and lacks numerical support. Consider providing surface roughness measurements or pore size distribution statistics to support the observation of roughness and density changes.

Uniform Distribution of d-UiO (Lines 134-138)

·       The claim of uniform dispersion needs quantitative verification. Is there any statistical analysis of uniformity, such as EDS mapping intensity variations or particle size distribution analysis?

Function of d-UiO in the Functional Layer (Lines 139-141)

·       "Moreover, due to the defective active sites……….." The sentence structure is unclear. It should be reworded for clarity. Also, explain how the increased defects contribute to membrane performance in terms of filtration, adsorption, or mechanical properties.

Defect Modification Verification (Lines 145-150)

·       "The SEM images of UiO-66-NH2 and d-UiO-3……….” Provide a more detailed explanation of how defect modification was confirmed. Are there quantitative metrics supporting this conclusion?

SBET and Microporosity Increase (Lines 161-167)

·       "Compared with UiO-66-NH2, the SBET of d-UiO-4 increased by 56%........." The reason for the SBET increase should be better supported with pore volume data or a specific discussion on how etching alters the framework structure. Additionally, how does this increase in surface area translate to membrane performance?

XRD Peak Analysis (Lines 178-185)

·       "However, the crystallization peak strength………." The term "better crystallinity" is vague. Does higher crystallization peak intensity always correlate with improved membrane properties? Please discuss how crystallinity impacts membrane performance.

FT-IR Analysis Interpretation (Lines 189-194)

·       "d-UiO-4@TUT-PVDF has C=O vibration at 1654..............." The functional group assignments should be referenced to relevant literature. Also, how does this spectral evidence support successful integration into the membrane?

XPS Confirmation of d-UiO Loading (Lines 202-205)

·       "d-UIO-4@TUT-PVDF shows strong peaks of F1s (688 eV), O1s (530 eV) and N1s (399 eV)..." While the presence of these elements supports membrane composition, can you provide quantitative atomic percentages to confirm successful functionalization?

Abstract and Conclusion

General Improvements:

·       Ensure a smooth transition from the problem statement to the research motivation, methodology, and key findings.

·       Explicitly highlight how this work advances COâ‚‚ separation compared to previous studies.

·       Some sentences are unclear due to awkward phrasing. Refining the structure will enhance readability.

·       Please also include at least one sentence to highlight the limitations and future research directions

Author Response

Reviewer 2:

General Comments

Comment 1: Improve logical flow by ensuring smooth transitions from challenges to the motivation behind the study.

Response 1: Thank you for pointing this out. We have sorted out the article to ensure a smooth transition in this part.

Comment 2: Clearly define the research gap by stating what is missing in current studies and how this work addresses those gaps.

Response 2: Thank you for pointing this out. We have added the sentences, such as “High performance membrane materials can be synthesized by different preparation strategies, the membrane with multilayer structure should be studied emphatically.”

Comment 3: Strengthen the introduction by explicitly outlining the expected contributions, particularly how the proposed membrane design outperforms previous designs.

Response 3: Thank you for pointing this out. We believe this work is valuable, we innovative design of this multi-layer structure enables precise control over thickness by utilizing various types of MOFs and assembly concentrations.  Additionally, it enhances the separation efficiency of complex gas mixtures through optimized layer selection, better than original PVDF membrane.

Comment 4: Define key technical terms to make novel synthesis strategies accessible to non-specialist readers.

Response 4: Thank you for pointing this out. We defined key technical terms, such as TA for Tannic acid, TFN for thin film nanocomposite, which make it accessible to non-specialist readers.

Comment 5: Provide purity details for all chemicals, especially critical reagents like ZrCl4, NHâ‚‚-BDC, and Ti-BALDH, as their quality can significantly impact synthesis results.

Response 5: Thank you for pointing this out. It is crucial to provide purity details for chemicals, we have added them in the text.

Comment 6: Ensure crucial experimental details for reproducibility, including stirring speed (rpm) during dissolution steps, cooling rate after solvothermal reactions, and centrifugation parameters.

Response 6: Thank you for pointing this out. We have added experimental details in the text.

Comment 7: Specify concentrations for TA, Ti-BALDH, and methanol solutions used in membrane modification.

Response 7: Thank you for pointing this out. We used 5% TA, 10% Ti-BALDH and 30 mL methanol solutions in the process.

Comment 8: Provide washing conditions to ensure reproducibility.

Response 8: Thank you for pointing this out. We added the washing conditions (such as room temperature, wash with ethanol, et al.) for membrane to ensure reproducibility.

Comment 9: Maintain consistency in chemical symbols and formatting (e.g., NHâ‚‚-BDC instead of NH2-BDC).

Response 9: Thank you for pointing this out. We have revised the chemical symbols and formatting.

Comment 10: Clearly indicate whether reagent ratios are in molar terms to avoid ambiguity.

Response 10: Thank you for pointing this out. We have indicated that the reagent ratio is in moles.

Comment 11: Include more quantitative comparisons, such as tabulated data for particle sizes and defect density.

Response 11: Thank you for pointing this out. We have set the comparison table for pore structure, selectivity of membranes with other MOF-containing MMMs, which provide more quantitative comparisons.

Comment 12: Strengthen the connection between structural/morphological changes and functional membrane performance, including permeability, antifouling behavior, and mechanical properties.

Response 12: Thank you for pointing this out. Gas permeation activation energy and mechanical properties are the focus of this study, and we have made an analysis of this part in the text.

Comment 13: Revise unclear sentences for better clarity and readability, ensuring the manuscript is polished and professional.

Response 13: Thank you for pointing this out. We have refined ambiguous sentences to enhance clarity and readability, thereby ensuring the professionalism of the article.

Introduction

Comment 14: "Petrochemical production, coal gasification………" the phrase "haze weather" is unclear. What does it mean?

Response 14: Thank you for pointing this out. We want to express the petrochemical production and coal gasification emission further aggravate the haze weather condition.

Comment 15: "Gas separation technology plays a key role ……" This is a broad statement. Specify which gas separation techniques are most effective or give examples and Mention how membrane separation compares to other techniques in terms of efficiency and cost.

Response 15: Thank you for pointing this out. We have modified the content of this part and explained that membrane separation technology has a series of advantages such as high efficiency and reduce energy consumption.

Comment 16: "Excellent membrane materials…….." Add an example of a material that meets these criteria to strengthen the argument.

Response 16: Thank you for pointing this out. We have added an example of a material that meets these criteria to strengthen the argument.

Comment 17: "Due to the high selectivity and large pore volume, MOF have……….” Please check the correct phrasing, it should be "MOFs have", not "MOF have". And "High selectivity" compared to what?

Response 17: Thank you for pointing this out. We have corrected the phrase. Besides, High selectivity is referred to “MOFs compare with other traditional adsorbents, such as activated carbon.”

Comment 18: "However, the limited interaction ……..” is there any study which tried to provide solution?

Response 18: Thank you for pointing this out. At present, highly selective MMMs can be prepared by modified MOFs, polymer modification, in-situ synthesis, crosslinking and other methods to meet different gas separation requirements.

Comment 19: (I think its relevant literature review?) The discussion on previous studies is informative but somewhat scattered. It would be helpful to structure this section better: First, discuss advances in polymer membranes. Then, introduce the role of MOFs in improving performance. Finally, highlight key challenges and gaps. Provide a brief summary of the key takeaways from these studies before introducing your research.

Response 19: Thank you for pointing this out. This suggestion is very helpful. We have made some changes to this part to ensure that it is more clear and complete in logic and expression, 

Comment 20: (I hope its author`s Novelty of the Work): "In this work, a biomatrix phenolic network coated……...." This section should be more explicit about the novelty of the approach. How is this different from previous MOF-MMM approaches? Also, Define "step assembly, alternating complexation, and cooperative coordination".

Response 20: Thank you for pointing this out. Firstly, titanium-tannic acid polydentate complex is formed on the surface of the film for pre-deposition, which provides the active site for d-UiO nanocrystals. Subsequently, d-UiO with different defect ratios was constructed using acetic acid as a competitive ligand and uniformly infiltrated onto the surface of the active film. The surface protection etched by tannic acid was performed on d-UiO. Moreover, the defect site of d-UiO increased the interaction between the layers and the titanium-tannic acid complex, and a stable MOFs layer was formed. Finally, by strengthening the coordination, the complex coating is constructed on the surface of the MOFs layer for secondary deposition capping, fixing the MOFs layer and enhancing the stability and functionality of the film.

Methodology

Materials Section (Lines 74-84):

Comment 21: Specify reagent purity, as this affects experimental outcomes. Additionally, gases (CHâ‚„, Nâ‚‚, COâ‚‚) should mention purity levels.

Response 21: Thank you for pointing this out. We have supplemented the purity of the reagents and gases used in the experimental section of the text.

UiO-66-NHâ‚‚ Synthesis (Lines 85-91)

Comment 22: The dissolution and stirring step needs clarification regarding stirring speed and temperature.

Response 22: Thank you for pointing this out. We have added stirring at room temperature at a speed of 1000 rpm.

Comment 23: The cooling process should specify whether it was natural cooling or controlled cooling.

Response 23: Thank you for pointing this out. The cooling process is natural cooling.

Comment 24: The ethanol washing step should include the volume per washing cycle and the number of cycles.

Response 24: Thank you for pointing this out. We have supplemented the volume of ethanol used and the number of washes in this section.

Defective UiO-66-NHâ‚‚ Synthesis (Lines 92-98)

Comment 25: The phrase "stirred at room temperature for 10 min" should specify the stirring speed (e.g., 500 rpm).

Response 25: Thank you for pointing this out. ZrCl4 (1 mmol) and NH2-BDC (1 mmol) were dissolved in 30 mL of DMF and stirred with 1000 rpm at room temperature for 10 min.

Comment 26: It is unclear whether the reaction time and temperature were optimized or based on previous studies.

Response 26: Thank you for pointing this out. The reaction time and temperature are optimized in order to make the reaction more complete and expect better results.

Comment 27: Did you perform control experiments to confirm the defect formation? Consider adding references to confirm that the defect structure was obtained.

Response 27: Thank you for pointing this out. We have provided reference [28] to support the defect structure was obtained.

TFN Membrane Fabrication (Lines 99-109)

Comment 28: The concentration of TA and Ti-BALDH solutions should be mentioned.

Response 28: Thank you for pointing this out. We used 5% TA, 10% Ti-BALDH solutions in the process.

Comment 29: Sufficient adsorption of TA" is vague. Provide adsorption equilibrium time or reference data.

Response 29: Thank you for pointing this out. We have cited the reference [26] to prove that 12 hours is sufficient for adsorption of TA.

Comment 30: The term "stored for at least 12 h" should indicate storage conditions (temperature, humidity).

Response 30: Thank you for pointing this out. We have added the storage conditions in this section.

Characterization Section (Lines 113-128)

Comment 31: The SEM preparation method should specify the working voltage and magnification used for imaging.

Response 31: Thank you for pointing this out. The operating voltage is 5 kV and the magnification is 10000.

Comment 32: The XRD analysis should specify scan range and step size.

Response 32: Thank you for pointing this out. The crystalline structure was studied using X-ray diffraction (XRD, MAX2500, Rigku, λ= 0.15406 nm, collection range from 2° to 80°).

Comment 33:TGA should indicate heating rate and atmosphere.

Response 33: Thank you for pointing this out. Thermogravimetric analysis (TGA) was performed in air atmosphere with rate of 5℃·min-1.

Results and Discussion

Clarity of SEM Image Descriptions (Lines 132-138)

Comment 34: The description of SEM images is somewhat qualitative and lacks numerical support. Consider providing surface roughness measurements or pore size distribution statistics to support the observation of roughness and density changes.

Response 34: Thank you for pointing this out. The SEM images can clearly observe the morphological characteristics of the material. Combined with the specific surface area test, we combine the images and data for analysis, and the pore structure data is displayed in the table.

Uniform Distribution of d-UiO (Lines 134-138)

Comment 35: The claim of uniform dispersion needs quantitative verification. Is there any statistical analysis of uniformity, such as EDS mapping intensity variations or particle size distribution analysis?

Response 35: Thank you for pointing this out. By comparing the original PVDF and observing the distribution of elements in SEM images and EDS, we believe that this conclusion holds, and in addition, it is a valuable suggestion that we will consider using this method for further characterization of the material.

Function of d-UiO in the Functional Layer (Lines 139-141)

Comment 36: "Moreover, due to the defective active sites……….." The sentence structure is unclear. It should be reworded for clarity. Also, explain how the increased defects contribute to membrane performance in terms of filtration, adsorption, or mechanical properties.

Response 36: Thank you for pointing this out. We have adjusted this sentence to make the sentence structure clear and complete, and explain how defects increase adsorption properties. Additional cavities are created, increasing the microporous structure of the material, and more unsaturated metal adsorption sites are formed.

Defect Modification Verification (Lines 145-150)

Comment 37:"The SEM images of UiO-66-NH2 and d-UiO-3……….” Provide a more detailed explanation of how defect modification was confirmed. Are there quantitative metrics supporting this conclusion?

Response 37: Thank you for pointing this out. We have provided reference [27] to support the defect modification was confirmed.

SBET and Microporosity Increase (Lines 161-167)

Comment 38: "Compared with UiO-66-NH2, the SBET of d-UiO-4 increased by 56%........." The reason for the SBET increase should be better supported with pore volume data or a specific discussion on how etching alters the framework structure. Additionally, how does this increase in surface area translate to membrane performance?

Response 38: We have added data to this section to better show the performance improvement. For example, “Compared with UiO-66-NH2 (671 m2·g-1, 0.25cm3·g-1), the SBET of d-UiO-4 (1048 m2·g-1, 0.40cm3·g-1) increased by 56%......”. Besides, the defect of d-UiO nanocrystals in the high-performance d-UiO@TUT-PVDF not only enhances the adsorption of CO2, but also accelerates the diffusion of CO2 through the defect cavity, which significantly improves the separation performance.

XRD Peak Analysis (Lines 178-185)

Comment 39: "However, the crystallization peak strength………." The term "better crystallinity" is vague. Does higher crystallization peak intensity always correlate with improved membrane properties? Please discuss how crystallinity impacts membrane performance.

Response 39: Thank you for pointing this out. The higher the degree of crystallinity, the higher the tensile strength, elastic modulus and hardness of the film generally. This is because the molecular chains in the crystalline region are arranged neatly and tightly, and the intermolecular forces are enhanced, so that the material can better resist deformation when it is stressed.

FT-IR Analysis Interpretation (Lines 189-194)

Comment 40: "d-UiO-4@TUT-PVDF has C=O vibration at 1654..............." The functional group assignments should be referenced to relevant literature. Also, how does this spectral evidence support successful integration into the membrane?

Response 40: Thank you for pointing this out. We have added references [29] to support these conclusions. The characteristic absorption peak was detected by FT-IR, if there do not appeared new absorption peak, indicating that they were successfully integrated into the membrane through force combination.

XPS Confirmation of d-UiO Loading (Lines 202-205)

Comment 41: "d-UIO-4@TUT-PVDF shows strong peaks of F1s (688 eV), O1s (530 eV) and N1s (399 eV)..." While the presence of these elements supports membrane composition, can you provide quantitative atomic percentages to confirm successful functionalization?

Response 41: Thank you for pointing this out. This is a very helpful advice, we believe that the presence of these elements supports membrane composition, we will consider a more specific quantitative analysis in the follow-up in-depth study.

Abstract and Conclusion

General Improvements:

Comment 42: Ensure a smooth transition from the problem statement to the research motivation, methodology, and key findings.

Response 42: Thank you for pointing this out. We have checked this section to make sure they were in the right logical order and easy to understand.

Comment 43: Explicitly highlight how this work advances COâ‚‚ separation compared to previous studies.

Response 43: Thank you for pointing this out. The defects of d-UiO nanocrystals not only enhance the adsorption of CO2, but also accelerate the diffusion of CO2 through the defect cavity, which significantly improves the separation performance. Obviously, what makes this work special is the use of defect strategies.

Comment 44: Some sentences are unclear due to awkward phrasing. Refining the structure will enhance readability.

Response 44: Thank you for pointing this out. We have revised these sentences to make the article more coherent.

Comment 45: Please also include at least one sentence to highlight the limitations and future research directions

Response 45: Thank you for pointing this out. Gas separation membrane technology has broad prospects in the field of energy saving and environmental protection and industrial gas purification, and it is necessary to continue to break through the performance of membrane materials in the future, and pay attention to the feasibility and economy of its industrial applications to promote the development of gas separation membranes.

Round 2

Reviewer 1 Report

Comments and Suggestions for Authors

Comments:

This revision has not yet reached the standard for publication. The manuscript still lacks detailed and coherent descriptions, as well as critical analyses to support the authors' arguments. The readability remains problematic, making it difficult for readers to follow the research narrative. A major revision is required before considering publication.

  1. Section 2 – PVDF as Precursor:
    The authors have added detailed information about PVDF as the precursor. However, the manuscript still lacks a description of the PVDF membrane preparation process. The authors should cite the relevant literature or describe the process explicitly, regardless of whether it is self-developed or adopted from previous studies.
  2. Section 2.3 – Molar Ratio of Acetic Acid:
    The explanation regarding the molar ratio of acetic acid needs to be restructured for clarity. It is recommended to:
    • First, define the fixed molar ratio of UiO-66-NH2.
    • Then, explain the purpose and reasoning behind adding acetic acid, using clear keywords such as:
      “… the ratio of acetic acid/Zr = % to obtain the …”.
    • Finally, clearly state the sample naming convention based on the acetic acid/Zr molar ratio. This restructuring will enhance readability and logical flow.
  3. Section 2.4 – Description of Preparation Process & Figures:
    The description in this section requires significant improvement to ensure clarity and consistency with the corresponding figures. Several issues need to be addressed:
    • Redundant and unclear descriptions: The authors do not need to repeat the preparation steps. The text should be concise and directly reference the figures.
    • PVDF membrane modification process inconsistency:
      1. The manuscript states that PVDF undergoes sequential immersion in TA and Ti-BALDH, but the final description only mentions a single integrated TA-TiIV solution. This contradiction should be clarified.
      2. The manuscript lacks clear labels for individual figures, making it difficult to distinguish between solution composition, metal ions, and the final membrane product. The authors should explicitly indicate these elements in the figure captions.
    • Unrelated content & duplication: The upper part of figure 1 appears to discuss the MOF defect structure and functionalization, which is unrelated to this section and overlaps with Figure 10h. The content should be reorganized to avoid redundancy.
    • Sample naming inconsistencies: The sample names are confusing and inconsistent throughout the text and figures. Specifically:
      • The manuscript states that TTN@PVDF is obtained first, followed by TUT-UiO@PVDF. However, after an additional process, the sample name changes to TUT-UiO-TTN@PVDF, while in the figure it appears as d-UiO@TUT-PVDF.
      • The authors must carefully standardize and consistently use sample names throughout the text and figures.

4.     Sandwich-like Structure in Figure 1:

  • The layer composition of the final sample in the sandwich-like structure shown in Figure 1 is unclear. The figure does not provide sufficient details on the identity of each layer.
  • Additionally, the manuscript does not elaborate on the sandwich-like structure in the main text beyond its mention in the title. The authors should include a clear explanation of:
    • The individual layers in the sandwich-like structure.
    • Their functional roles in the material system.
    • How this structure correlates with the experimental results.
  1. SEM Analysis – d-UiO Layer Thickness & Deposition Verification:
    • The authors mention the thickness of the d-UiO layer on the membrane surface. However, this thickness should be quantitatively verified in both the text and corresponding figure.
    • Additionally, the TTN@PVDF sample should be explicitly presented and compared in the SEM analysis to confirm whether the d-UiO layer is deposited only on the surface or penetrates into the internal pores.

Author Response

This revision has not yet reached the standard for publication. The manuscript still lacks detailed and coherent descriptions, as well as critical analyses to support the authors' arguments. The readability remains problematic, making it difficult for readers to follow the research narrative. A major revision is required before considering publication.

1. Section 2 – PVDF as Precursor:
The authors have added detailed information about PVDF as the precursor. However, the manuscript still lacks a description of the PVDF membrane preparation process. The authors should cite the relevant literature or describe the process explicitly, regardless of whether it is self-developed or adopted from previous studies.

Response: Thank you for pointing this out. We have added the relevant literature [33-34] to the content of this part.

2. Section 2.3 – Molar Ratio of Acetic Acid:
The explanation regarding the molar ratio of acetic acid needs to be restructured for clarity. It is recommended to:

    • First, define the fixed molar ratio of UiO-66-NH2.
    • Then, explain the purpose and reasoning behind adding acetic acid, using clear keywords such as:
      “… the ratio of acetic acid/Zr = % to obtain the …”.
    • Finally, clearly state the sample naming convention based on the acetic acid/Zr molar ratio. This restructuring will enhance readability and logical flow.

Response: Thank you for pointing this out. We defined the fixed molar ratio of UiO-66-NH2. Besides, acetic acid was added to induce the material to form more defect sites. In addition, we made the naming of the sample specific. “According to the different molar ratio of acetic, d-UiO-1(1:15), d-UiO-2(1:30), d-UiO-3(1:70) and d-UiO-4(1:100) were obtained.”

3. Section 2.4 – Description of Preparation Process & Figures:
The description in this section requires significant improvement to ensure clarity and consistency with the corresponding figures. Several issues need to be addressed:

    • Redundant and unclear descriptions: The authors do not need to repeat the preparation steps. The text should be concise and directly reference the figures.

Response: Thank you for pointing this out. We have modified this part of the content to remove unnecessary redundancy.

    • PVDF membrane modification process inconsistency:
    • The manuscript states that PVDF undergoes sequential immersion in TA and Ti-BALDH, but the final description only mentions a single integrated TA-TiIV solution. This contradiction should be clarified.

Response: Thank you for pointing this out. We have revised the content “immersed in TA and Ti-BALDH solution.”

The manuscript lacks clear labels for individual figures, making it difficult to distinguish between solution composition, metal ions, and the final membrane product. The authors should explicitly indicate these elements in the figure captions.

    • Unrelated content & duplication: The upper part of figure 1 appears to discuss the MOF defect structure and functionalization, which is unrelated to this section and overlaps with Figure 10h. The content should be reorganized to avoid redundancy.

Response: Thank you for pointing this out. We have made modifications and adjustments to Figure 1, indicating these elements in the figure captions explicitly.

    • Sample naming inconsistencies: The sample names are confusing and inconsistent throughout the text and figures. Specifically:
      • The manuscript states that TTN@PVDF is obtained first, followed by TUT-UiO@PVDF. However, after an additional process, the sample name changes to TUT-UiO-TTN@PVDF, while in the figure it appears as d-UiO@TUT-PVDF.
      • The authors must carefully standardize and consistently use sample names throughout the text and figures.

Response: Thank you for pointing this out. We have revised the sample name to TUT-UiO-1-TTN@PVDF, TUT-UiO-2-TTN@PVDF, TUT-UiO-3-TTN@PVDF, TUT-UiO-4-TTN@PVDF.

4. Sandwich-like Structure in Figure 1:

  • The layer composition of the final sample in the sandwich-like structure shown in Figure 1 is unclear. The figure does not provide sufficient details on the identity of each layer.
  • Additionally, the manuscript does not elaborate on the sandwich-like structure in the main text beyond its mention in the title. The authors should include a clear explanation of:
    • The individual layers in the sandwich-like structure.
    • Their functional roles in the material system.
    • How this structure correlates with the experimental results.

Response: Thank you for pointing this out. We have modified the Figure 1 and added some captions to make it easier to understand. In addition, we elaborated on the sandwich-like structure in the main text.

5. SEM Analysis – d-UiO Layer Thickness & Deposition Verification:

    • The authors mention the thickness of the d-UiO layer on the membrane surface. However, this thickness should be quantitatively verified in both the text and corresponding figure.
    • Additionally, the TTN@PVDF sample should be explicitly presented and compared in the SEM analysis to confirm whether the d-UiO layer is deposited only on the surface or penetrates into the internal pores.

Response: Thank you for pointing this out. We have checked this content, SEM image (f) showed the thickness of the d-UiO layer on the membrane surface(~100 nm). Besides, we confirmed the d-UiO penetrates into the internal pores from cross-section of SEM image (g).

Again, we really thank you  for the insightful comments.

Reviewer 2 Report

Comments and Suggestions for Authors

Dear Authors,

Thank you for the revised version. After reading this revised manuscript, I still have some concerns as follows:

Providing specific examples and references instead of generic statements. Please check the manuscript thoroughly to avoid the redundancy, for example:

Repetition 
 "During the traditional industrial separation process, approximately 45-55% of total energy consumption is attributed to this step. Membranebased separation technology can reduce energy consumption by up to 90% compared to heat-driven distillation methods. Among these, membrane technology for gas separation has shown great prospect after technological breakthroughs[5]." 

No referece?
"Consequently, it has found extensive application in areas such as oily wastewater treatment, water purification, food processing, gas separation, and other practical industrial applications."

"High performance membrane materials can be synthesized by different preparation strategies, the membrane with multilayer structure should be studied emphatically. However, the concurrent optimization of high permeability and high selectivity remains a central challenge in the field of polymer membranes for gas separation."

"The gas permeability coefficient was deter- 150 mined by constant pressure and variable volume method, the sample is fixed with the gas 151 passes through with the test temperature and pressure of 25℃ and 100 kPa, . Repeated 152 measurement three times to take the average, the gas flow measured by the flowmeter."

Similarity 25%? Reduce it to the below 15%

Novelty

Comment 20 (Novelty and Definitions) – The response explains the approach well but lacks clarity on how it differs from previous MOF-MMM approaches. Additionally, "step assembly, alternating complexation, and cooperative coordination" should be explicitly defined.

Comment 8 (Washing Conditions) – Please also add the time for washing durations.

Please check throughout the manuscript.......

Good luck.

Author Response

Dear Authors,

Thank you for the revised version. After reading this revised manuscript, I still have some concerns as follows:

Providing specific examples and references instead of generic statements. Please check the manuscript thoroughly to avoid the redundancy, for example:

Repetition
"During the traditional industrial separation process, approximately 45-55% of total energy consumption is attributed to this step. Membrane-based separation technology can reduce energy consumption by up to 90% compared to heat-driven distillation methods. Among these, membrane technology for gas separation has shown great prospect after technological breakthroughs [5]." 

Response: Thank you for pointing this out. We have revised this part of the content. “Membrane-based separation technology can reduce energy consumption by up to 90% compared to heat-driven distillation methods, which has shown great prospect after technological breakthroughs [5].”

No reference?
"Consequently, it has found extensive application in areas such as oily wastewater treatment, water purification, food processing, gas separation, and other practical industrial applications."

Response: Thank you for pointing this out. We have added references [10-14] to the content of this part.

"High performance membrane materials can be synthesized by different preparation strategies, the membrane with multilayer structure should be studied emphatically. However, the concurrent optimization of high permeability and high selectivity remains a central challenge in the field of polymer membranes for gas separation."

Response: Thank you for pointing this out. We have added references [31-32] to the content of this part.

"The gas permeability coefficient was determined by constant pressure and variable volume metho, the sample is fixed with the gas passes through with the test temperature and pressure of 25℃ and 100 kPa. Repeated measurement three times to take the average, the gas flow measured by the flowmeter."

Response: Thank you for pointing this out. We have added references [34] to the content of this part.
Similarity 25%? Reduce it to the below 15%

Response: Thank you for pointing this out. We have checked and revised the article.

Novelty
Comment 20 (Novelty and Definitions) – The response explains the approach well but lacks clarity on how it differs from previous MOF-MMM approaches. Additionally, "step assembly, alternating complexation, and cooperative coordination" should be explicitly defined.

Response: Thank you for pointing this out. A coating that encapsulates MOF nanocrystals is constructed on the membrane surface. The MOF nanocrystal layer is immobilized through the coordination cross-linking between TiIV ions and the polyphenol tannic acid (TA), which effectively enhances the gas separation performance of the membrane. Compared with traditional MOF-MMM approaches, this innovative design of the multilayer structure not only allows for the adjustment of the membrane thickness by changing the type and assembly concentration of MOF nanocrystals, but also improves the separation efficiency of complex gas mixtures through the optimization of the selective layer.

Comment 8 (Washing Conditions) – Please also add the time for washing durations.
Response: Thank you for pointing this out. We have added the time for washing durations.
Again, we really thank you and the reviewers for the insightful comments.

Please check throughout the manuscript.......

Good luck.

Round 3

Reviewer 1 Report

Comments and Suggestions for Authors

I have reviewed the changes made by the authors and am pleased to confirm that all previous concerns and suggestions have been adequately addressed. The revisions have significantly improved the clarity and quality of the manuscript. It can be accepted for publication in Membranes.

Reviewer 2 Report

Comments and Suggestions for Authors

Dear authors,

Thank you for revision and addressing the comments which are acceptable. However, similarity is still 25%.